# Effects of non-pelleted or pelleted low-native grass and pelleted high-native grass diets on meat quality by regulating the rumen microbiota in lambs

Tingyu Liu,[1] Zhenkun Bu,[2,3] Kaifeng Xiang,[4] Yushan Jia,[5] Shuai Du[5]

**ABSTRACT** Diet modulates the rumen microbiota, which in turn can impact the animal performance. The rumen microbiota is increasingly recognized for its crucial role in regulating the growth and meat quality of the host. Nevertheless, the mechanism by which the rumen microbiome influences the fatty acid and amino acid profiles of lambs in the grass feeding system remains unclear. This study aimed to evaluate the effects of different native grass-based diets on animal performance, meat quality, fatty acid compositions, amino acid profiles, and rumen microbiota of lamb. Seventy-two Ujumqin lambs were randomly assigned into three treatments according to the initial body weight ($27.39 \pm 0.51$ kg) and age (6 months $\pm$ 6 days). The lambs received three diets: (i) non-pelleted native grass hay with 40% concentrate diet; the native grass and concentrate were fed individually; (ii) pelleted native grass hay with 40% concentrate diet (PHLC); (iii) pelleted native grass hay with 60% concentrate diet (PHHC). The results showed that among the three groups, the PHHC and PHLC diets had markedly ($P < 0.05$) higher average daily gain and $pH_{45\ min}$, respectively. All amino acid levels were significantly ($P < 0.05$) decreased in the PHHC diet than in the PHLC diet. The principal coordinate analysis of the ruminal microbiota indicated the markedly distinct separation ($P = 0.001$) among the three groups. In addition, the correlation analysis showed that the *Rikenellaceae*_RC9_gut_group, *Prevotellaceae*_UCG-003, *Succinivibrio*, and *Succiniclasticum* were significantly ($P < 0.05$) associated with most of the fatty acid and amino acid profiles. The correlation analysis of the association of microbiome with the meat quality provides us with a comprehensive understanding of the composition and function of the rumen microbial community, and these findings will contribute to the direction of future research in lamb.

**IMPORTANCE** Diet modulates the gut microbiome, which in turn impact the meat quality, yet few studies investigate the correlation between the rumen microbiome and the fatty acid profile of meat. Here, the current study develops an experiment to investigate the correlation of the rumen microbiome and fatty acid profile of meat: rumen microbiome responses to feed type and meat quality. The results indicated a unique microbiota in the rumen of lamb in response to diets and meat quality. Associations between utilization and production were widely identified among the affected microbiome and meat quality, and these findings will contribute to the direction of future research in lamb.

**KEYWORDS** hay, concentrate, pelleted diet, lamb, meat quality, rumen microbiota

Ruminants have evolved over many years and are one of the most widely consumed meats for humans. In recent years, the demand of high-quality and healthy ruminant meat has been increasing. Hence, an effective practice for improving the meat quality

Address correspondence to Shuai Du, dushuai_nm@sina.com.

The authors declare no conflict of interest.

See the funding table on p. 15.

should be developed. Ruminants cannot provide an abundant source of the animal protein products to meet the nutrition demand of the growing population worldwide (1) but are a major source of wool and represent a distinct class of animals because it has a specialized digestive organ, the rumen, which ferments the feed and converts fiber-rich plant materials and nonhuman edible plant materials to protein via the microbial community (2). The rumen leads to the development of mutualistic symbiosis between hosts and rumen microbial community composition, which could provide about 70% energy for the ruminant needs (3). Additionally, the rumen microbiota has been linked to host feed efficiency and animal performance (4).

Traditionally, the lambs were thinner and weaker (even leading to death) when fed with native grass hay in winter and spring (5). The native grass pellets or non-pelleted native grass hay supplements with concentrate diets have become common by increasing the dry matter intake (DMI) and the average daily gain (ADG) in sheep production systems (6). Previously published research found that bacteria play an important role in most of the feed biopolymer degradation and fermentation, which indicated that bacteria are key players for animal performance (1). Nevertheless, the rumen microbiota is strongly influenced by individual genetics, animal age, feed type, and feeding system and especially directly linked to diet (7). On the one hand, the low-forage diets could affect the composition of the rumen microbiota (8), and the high-forage diets are beneficial for the genus *Firmicutes* (9). On the other hand, previous reports also found that the pellet diets could reduce the bacterial richness in the rumen (10), whereas others found that the pelleted-hay diets had a greater increase in bacterial richness (11). Compared to the native grass pellet supplements with concentrate diets, the pelleted native grass hay could increase the animal performance and the bacterial abundance in the rumen (12). The alternation of rumen microbiota in turn influences the meat quality via the intramuscular fatty acid and amino acid metabolism, such as *Prevotella*, *Clostridiales*, and *Ruminococcaceae*, which is likely related to lipid and protein metabolism (13–15). However, to our knowledge, little information is available on the effect of the non-pelleted or pelleted native grass diets on the meat quality, fatty acid and amino acid profiles, composition and function of the rumen microbiota.

In the present study, the author hypothesized that non-pelleted native grass hay with 40% concentrate diet (NPH), pelleted native grass hay with 40% concentrate diet (PHLC), and pelleted native grass hay with 60% concentrate diet (PHHC) could affect the animal performance, meat quality, fatty acid compositions, amino acid profiles, and rumen microbiota in lamb. Therefore, this study aimed to assess the effects of the NPH, PHLC, and PHHC diets on growth performance, meat quality, fatty acid compositions, amino acid profiles, and rumen microbiota of lamb. In addition, the correlations between the rumen microbiota and fatty acid or amino acid profiles were also investigated.

## MATERIALS AND METHODS

### Animals, experimental design, and feeding management

Seventy-two Ujumqin lambs were randomly assigned into three treatments according to the initial body weight (BW, $27.39 \pm 0.51$ kg) and age (6 months $\pm$ 6 days). The lambs were received three diets: (i) non-pelleted native grass hay with 40% concentrate diet; the native grass and concentrate were fed individually, (ii) pelleted native grass hay with 40% concentrate diet; (iii) pelleted native grass hay with 60% concentrate diet. A pellet machine (H.S. 508 Pellets Mill; Liyang Weifang Equipment Co., Ltd., Liyang, China) was used to produce pellets. The concentrate was purchased from a local company (Ruiyuan Agriculture and Animal Husbandry Co., Ltd., Xilinhot, China) and comprised of 25% maize, 8% soybean meal, 4% distillers dried grains with solubles, and 3% wheat bran. The metabolizable energy was calculated according to the previously published method of the following formula: ME = GE − FE − UE − Eg (16).

The feeding experiment was carried out at the Lvye Grass-based Livestock Husbandry Development Co., Ltd. (Xilin Hot, China). Lambs were randomly blocked into 18 units, six

units per treatment and four lambs per cage in a completely randomized design based on similar BW before morning feeding. The native grass was from the typical steppe in Xilinhot, Inner Mongolian Plateau. *Stipa gigantea* L. and *Leymus chinensis* (Trin.) Tzvel. are the dominant species in the typical steppe; additionally, *Lespedeza davurica* (Laxm.) Schindl, *Allium mongolicum* Regel, *Thalictrum petaloideum* Linn., *Bupleurum chinensis* DC., *Serratula centauroides* Linn., *Caragana microphylla* Lam, and others were also found in the typical steppe. The feeding experiment lasted for 75 days, with the first 15 days for adaptation period for the diet and environment and the next 60 days for sampling and data collection. The ingredient structure and chemical compositions of the experimental diets are listed in Table 1.

## Feed intake and growth performance

The experimental diets were provided twice daily, early morning (08:00) and late afternoon (16:00), at a rate of 110% of *ad libitum* intake calculated by weighing back refusals daily. The water was provided *ad libitum*. During the whole experiment, all lambs were weighed before the morning feeding with an empty stomach and without fasting throughout the experimental period in the morning (06:00–07:00) and with a week interval; the initial and final BW were also recorded to evaluated the ADG.

## Physical meat quality analysis

All lambs were weighed on day 75 and fasted for 12 h before slaughtering. One lamb from each cage was randomly selected, and a total of 18 lambs were slaughtered at a commercial abattoir for carcass measurements and meat quality assessment when the experiment was finished. After slaughtering, the hot carcass weight of each lamb was recorded to calculate carcass yield. Subsequently, about 500 g of the longissimus dorsi muscle was taken from the left side of the carcass, placed in self-sealed bags, and then stored at 4°C for analysis of meat quality and meat nutritional value. The dressing percentage was determined by dividing the carcass weight by the live weight (17). The bones and meat were separated from the carcass, and the net meat percentage was calculated according to the net meat mass: total carcass weight ratio × 100% (5). The backfat thickness was determined by measuring the tissue thickness between the 12th and 13th ribs. The moisture, protein, and intramuscular fat of the carcass from the longissimus dorsi muscle was measured according to the Association of Official Analytical Chemists methods (18). The pH value of the longissimus dorsi muscle was measured at 45 min and 24 h with a portable pH meter (STARTED 100/B; Ohaus,

**TABLE 1** Ingredients and chemical composition of the diets[b]

| Item | NPH | PHLC | PHHC |
|---|---|---|---|
| Ingredients (% DM) | | | |
| Native grass | 60.00 | 60.00 | 40.00 |
| Concentrate | 39.96 | 39.96 | 59.96 |
| Salt | 0.02 | 0.02 | 0.02 |
| Mineral premix[a] | 0.02 | 0.02 | 0.02 |
| Chemical compositions | | | |
| Dry matter (%) | 92.31 | 91.84 | 92.43 |
| Organic matter (% DM) | 95.67 | 95.28 | 95.71 |
| Crude protein (% DM) | 9.79 | 9.92 | 10.08 |
| Ether extract (% DM) | 2.28 | 2.91 | 3.12 |
| Acid detergent fiber (% DM) | 39.27 | 39.04 | 37.04 |
| Neutral detergent fiber (% DM) | 55.47 | 56.02 | 54.31 |
| Metabolizable energy (MJ/kg DM) | 9.54 | 9.62 | 10.04 |

[a]Composition of mineral premix. Per kilogram: magnesium 1600 mg; cuprum 150 mg; ferrum 400 mg; zinc 450 mg; cobalt 5 mg; selenium 15 mg; iodine 60 mg.
[b]DM, dry matter; NPH, non-pelleted native grass hay with 40% concentrate diet; PHLC, pelleted native grass hay with 40% concentrate diet; PHHC, pelleted native grass hay with 60% concentrate diet. Mineral have calculated values based on commercial products.

Shanghai, China). The lightness (L*), redness (a*), and yellowness (b*) were evaluated by a spectrophotometer (Minolta CM-2002, Osaka, Japan) (19). The cooking yield was calculated as the percent weight loss relative to the initial sample weight (5). The shear force of the longissimus dorsi muscle was determined by a C-LM tenderness tester (Tenovo International Co., Limited, Beijing, China) based on the manufacturer's instructions. The fatty acid profiles were measured according to the published method with a gas chromatography–mass spectrometer 7890B (Agilent, California, USA) (16). The samples were melted in a steam bath or oven at 10°C above the melting point. If the melted fat was cloudy, it was filtered through filter paper. Methyl esters of fatty acids were prepared from 400- to 500-mg fat. Before infrared analysis, excess impurities were removed with a suitable cleanup procedure. Then, the undiluted fatty acid methyl esters were weighed to the nearest 0.1 mg into a 25-mL volumetric flask. A cell was filled with $CS_2$ solution, and a matching cell was filled with a test sample. Finally, the test sample or calibration solution was scanned in the same range as that of the reference. The amino acid profiles were measured according to the published method with an automatic amino acid analyzer (L-8800, Hitachi Ltd, Tokyo, Japan) (20). Briefly, the samples were added to 15 mL of 6 mol/L HCl with three to four drops of phenol. After hydrolyzation at 110°C ± 1°C for 22 h under nitrogen, the samples were filtrated, and 1-mL supernatant was evaporated in a vacuum drying oven at 50°C and then redissolved in 1 mL of saline sodium citrate.

## Rumen sample collection, DNA extraction, and 16S rRNA gene amplification and sequencing

One lamb from each cage was randomly selected, and a total of 18 lambs were sampled for analyzing the rumen microbiome. Approximately 100 mL of rumen contents were collected and transported into sterile centrifuge tubes and immediately frozen in liquid nitrogen containers; then, the samples were stored at –80°C for future analysis. The E.Z.N.A. Stool DNAKit (D4015, Omega, Inc., USA) was used to extract the DNA of rumen samples by the method in the manufacturer's instructions. The NanoDrop 2000 UVvis Spectrophotometer (Thermo Scientific, Wilmington, USA) was used to determine the purity and concentration of the extracted DNA, and 1% agarose gel electrophoresis was used to assess the extracted DNA quality (21). The PCR amplification and bioinformatic analysis were performed in LC-Bio Technology Co., Ltd., (Hang Zhou, China). The variable regions V3–V4 of the bacterial 16S rDNA gene were amplified with primers 341F (F: 5′-CCTACGGGNGGCWGCAG-3′) and 805R (R: 5′-GACTACHVGGGTATCTAATCC-3′) (22). The amplification was conducted following the description of Tian et al. (23).

## Bioinformatics analysis

The low-quality reads (quality scores lower than 20) and short reads (lower than 100 bp) were trimmed by using the sliding window algorithm method in fqtrim (v 0.94). Quality fltering was performed to obtain high-quality clean tags according to fqtrim. Chimeric sequences were filtered using Vsearch software (v2.3.4). Editing, unique sequences selection, identification of chimeras, read assembly, and determination and taxonomic classification of amplicon sequence variants (ASVs) were made using Divisive Amplicon Denoising Algorithm in R (version 3.5.1) (24). The ASVs were classified into organisms with a naïve Bayesian model according to the RDP classifier (http://rdp.cme.msu.edu/) based on SILVA database (25), with the confidence threshold value of 90%. The Venn diagram was used to display the unique and common ASVs by R (version 1.6.2). The Chao1 value and Shannon index were used to evaluate the richness and diversity of alpha diversity for these samples (26). The Good's coverage was calculated through QIIME software. Principal coordinate analysis (PCoA) was conducted by R (version 3.3.1) software based on the Bray-Curtis distance metrics, and the ellipses represent the confidence in the fitting. The permutational multivariate analysis of variance (PERMA-NOVA) test was applied to analyze the significant difference among these treatments

with R (version 2.5.4) software (27). The false discovery rate-adjusted Kruskal–Wallis multiple comparisons ($P < 0.05$) were used to detect the ruminal microbiota at the phylum and genus levels (28). The main differentially abundant genera were analyzed by the linear discriminant analysis (LDA) coupled with effect size (LEfSe) method (29). The Pearson correlation analysis was used to determine the relationship between the ruminal microbiota and fatty acid or amino acid profiles using R (version 4.1.3).

## Statistical analysis

The intake, animal performance, carcass characteristics, meat quality, fatty acid, and amino acid profiles data were analyzed by SAS software (version 9.0) (SAS Inst., Inc., Cary, NC) as a completely randomized design per unit used as experimental units. Data were checked for normality using the PROC UNIVARIATE procedure of SAS (SAS Inst., Inc., Cary, NC). All data were calculated from measurements collected throughout the study and analyzed as repeated measures using the PROCMIXED procedure of SAS (SAS Inst., Inc., Cary, NC). Significant differences among groups were analyzed by a one-way analysis of variance according to the statistical model, $Y = \mu + \alpha + \varepsilon$, where Y = observation, $\mu$ = overall mean, $\alpha$ = diet effect, and $\varepsilon$ = error, and Duncan's tests, with $P < 0.05$ as statistical significance using SAS version 9.0 and the diets as fixed factors.

## RESULTS

### Effects of diets on animal performance and carcass characteristics

The results for growth performance and carcass characteristics are shown in Table 2. As expected, no significant ($P > 0.05$) difference was found among the three groups on the initial BW. The final BW was influenced by the diets, and the markedly ($P < 0.05$) higher final BW was observed in the PHHC group compared to that in the NPH and PHLC groups. The significantly ($P < 0.05$) higher ADG was also found in the PHHC group compared to that in the NPH and PHLC groups, whereas no significant ($P > 0.05$) difference was found in the DMI and carcass weight among the three groups. The BW before slaughter was significantly ($P < 0.05$) affected by the diet and with a tendency to be higher in the PHHC group than that in the NPH and PHLC groups. Interestingly, there was no significant ($P > 0.05$) difference in the dressing percentage, net meat mass, net meat percentage, and backfat among the three groups.

### Meat quality

The effects of diets on the chemical compositions and meat quality of lambs were shown in Table 3. The diets had no significant ($P > 0.05$) effects on the moisture of the meat. The intramuscular fat in the NPH and PHLC groups was statistically ($P < 0.05$) lower than

**TABLE 2**  Effects of the diets on growth performance and carcass characteristics of lamb[a]

| Item | NPH | PHLC | PHHC |
|---|---|---|---|
| Initial BW (kg) | 26.83 ± 0.33 | 27.17 ± 0.61 | 26.92 ± 0.30 |
| Final BW (kg) | 35.25 ± 0.94[b] | 34.83 ± 1.00[b] | 38.67 ± 0.95[a] |
| ADG (g/d) | 140.28 ± 14.66[b] | 127.78 ± 13.21[b] | 195.83 ± 16.91[a] |
| Dry matter intake | 1.62 ± 0.01 | 1.65 ± 0.01 | 1.65 ± 0.01 |
| Carcass weight (kg) | 18.17 ± 0.35 | 17.93 ± 0.43 | 18.85 ± 0.32 |
| BW before slaughter (kg) | 34.33 ± 0.97[b] | 32.97 ± 0.97[b] | 37.33 ± 0.64[a] |
| Dressing percentage (%) | 48.71 ± 0.70 | 47.33 ± 0.43 | 48.32 ± 0.46 |
| Net meat mass (kg) | 14.02 ± 0.32 | 14.02 ± 0.29 | 14.34 ± 0.43 |
| Net meat percentage (%) | 77.17 ± 0.81 | 78.18 ± 0.41 | 76.07 ± 1.00 |
| Backfat thickness (cm) | 0.63 ± 0.08 | 0.74 ± 0.08 | 0.61 ± 0.10 |

[a]Within a row, means without a common superscript letter (a, b) are different ($P < 0.05$). ADG, average daily gain; BW, body weight; NPH, non-pelleted native grass hay with 40% concentrate diet, PHLC, pelleted native grass hay with 40% concentrate diet; PHHC, pelleted native grass hay with 60% concentrate diet.

**TABLE 3** Effects of the diets on chemical compositions and meat quality of lamb[a]

| Item | NPH | PHLC | PHHC |
|---|---|---|---|
| Chemical compositions | | | |
| Moisture (g/100 g) | $74.53 \pm 0.59$ | $74.60 \pm 0.42$ | $73.17 \pm 0.95$ |
| Intramuscular fat (g/100 g) | $3.39 \pm 0.22^b$ | $3.64 \pm 0.17^b$ | $5.16 \pm 0.14^a$ |
| Protein (g/100 g) | $23.34 \pm 0.50^a$ | $22.01 \pm 0.23^b$ | $21.80 \pm 0.04^b$ |
| Meat quality | | | |
| $pH_{45\,min}$ | $6.56 \pm 0.09^a$ | $6.29 \pm 0.07^b$ | $6.54 \pm 0.05^a$ |
| $pH_{24\,h}$ | $5.51 \pm 0.09$ | $5.56 \pm 0.15$ | $5.40 \pm 0.15$ |
| $L^*$ | $29.91 \pm 0.49$ | $30.46 \pm 0.57$ | $30.99 \pm 0.22$ |
| $a^*$ | $14.94 \pm 0.18^b$ | $15.53 \pm 0.10^a$ | $15.51 \pm 0.14^a$ |
| $b^*$ | $4.37 \pm 0.21^a$ | $3.37 \pm 0.07^b$ | $3.32 \pm 0.03^b$ |
| Cooking yield (%) | $58.42 \pm 1.19$ | $56.20 \pm 3.49$ | $55.92 \pm 0.61$ |
| Shear force (N) | $94.78 \pm 0.86^c$ | $114.90 \pm 0.95^a$ | $103.08 \pm 3.47^b$ |

[a]Within a row, means without a common superscript letter (a, b, c) are different ($P < 0.05$). $L^*$, lightness; $a^*$, redness; $b^*$, yellowness. NPH, non-pelleted native grass hay with 40% concentrate diet, PHLC, pelleted native grass hay with 40% concentrate diet; PHHC, pelleted native grass hay with 60% concentrate diet.

that in the PHHC group. Interestingly, the results for protein were contrasting, and the markedly ($P < 0.05$) higher protein was observed in the NPH group than that in the PHLC and PHHC groups. Compared to the NPH and PHHC groups, the significantly ($P < 0.05$) higher $pH_{45\,min}$ was found in the PHLC group, and no significant ($P > 0.05$) difference was found on the $pH_{24\,h}$ among the three groups. Similarly, there was no significant ($P > 0.05$) difference on the $L^*$ in the three groups. The significantly ($P < 0.05$) higher $a^*$ was observed in animals fed with PHHC and PHLC diets, and markedly ($P < 0.05$) higher $b^*$ was found in the NPH group than that in the PHLC and PHHC groups. There was no significant ($P > 0.05$) difference on the cooking yield among the three groups. The highest shear force was detected in the PHLC group, followed by the parameter in the PHHC and NPH groups with significant ($P < 0.05$) difference.

## Fatty acid profiles in the longissimus dorsi muscle

The fatty acid profiles of lambs were greatly altered by the diets (Table 4). Compared to the NPH group, the PHLC group significantly ($P < 0.05$) decreased the concentration of C14:0, C14:1, C16:0, C16:1, C18:1n9t, C18:2n6t, C18:2n6c, C18:3n3, C24:0, C22:4n6, polyunsaturated fatty acid (PUFA), n-3 PUFA, and the PUFA:saturated fatty acid (SFA) ratio, whereas the concentrations of C14:0, C14:1, C16:0, C16:1, C18:1n9t, C18:2n6t, C18:2n6c, and PUFA were significantly ($P < 0.05$) increased in the PHHC group than those in the NPH group. The concentrations of C18:0 and C18:1n9c were followed by the order PHHC, PHLC, and NPH groups with significant ($P < 0.05$) difference. Additionally, the PHHC diet remarkably ($P < 0.05$) enhanced the concentration of SFA, monounsaturated fatty acid (MUFA), n-6 PUFA, and the n-6:n-3 PUFA ratio compared to those in the NPH group.

## Free amino acid profiles in the longissimus dorsi muscle

The influence of the diets on the free amino acid profiles of lambs is shown in Table 5. The PHHC diet significantly ($P < 0.05$) decreased all of the amino acid contents compared to those in the PHLC diet. Interestingly, no significant ($P > 0.05$) differences were found between the NPH and PHLC diets on all of the amino acid expect for the cystine, and the markedly ($P < 0.05$) higher cystine was detected in the NPH group. Meaningfully, the NPH and PHLC diets significantly ($P < 0.05$) enhanced the flavor amino acid content than that in the PHHC group.

## The ruminal microbiota

The rumen microbiota of all the NPH, PHLC, and PHHC treatments was analyzed by 16S rRNA sequencing (Table 6). A total of 1,521.250 raw reads were obtained for the 18 samples

**TABLE 4** Effects of the diets on fatty acid profiles of lamb[a]

| Item | NPH | PHLC | PHHC |
|---|---|---|---|
| C14:0 (g/100 g) | 8.87 ± 0.07[b] | 7.08 ± 0.06[c] | 14.95 ± 0.31[a] |
| C14:1 (g/100 g) | 0.80 ± 0.01[b] | 0.65 ± 0.02[c] | 1.04 ± 0.02[a] |
| C16:0 (g/100 g) | 85.93 ± 1.06[b] | 77.80 ± 0.86[c] | 142.45 ± 2.03[a] |
| C16:1 (g/100 g) | 6.19 ± 0.12[b] | 5.33 ± 0.16[c] | 9.09 ± 0.10[a] |
| C18:0 (g/100 g) | 58.93 ± 0.92[c] | 75.33 ± 0.88[b] | 113.85 ± 2.61[a] |
| C18:1n9t (g/100 g) | 8.28 ± 0.34[b] | 6.56 ± 0.09[c] | 13.80 ± 0.26[a] |
| C18:1n9c (g/100 g) | 122.13 ± 1.02[c] | 136.00 ± 2.22[b] | 208.50 ± 0.18[a] |
| C18:2n6t (g/100 g) | 1.13 ± 0.01[b] | 0.84 ± 0.02[c] | 1.45 ± 0.01[a] |
| C18:2n6c (g/100 g) | 15.50 ± 0.13[b] | 14.43 ± 0.55[c] | 18.05 ± 0.05[a] |
| C20:0 (g/100 g) | 1.14 ± 0.05[b] | 1.06 ± 0.05[b] | 1.35 ± 0.01[a] |
| C18:3n3 (g/100 g) | 2.59 ± 0.04[a] | 1.73 ± 0.03[c] | 2.04 ± 0.02[b] |
| C22:0 (g/100 g) | 1.43 ± 0.02 | 1.44 ± 0.02 | 1.49 ± 0.03 |
| C20:3n6 (g/100 g) | 3.84 ± 0.07[b] | 4.30 ± 0.07[a] | 4.27 ± 0.01[a] |
| C24:0 (g/100 g) | 3.59 ± 0.05[a] | 2.61 ± 0.02[c] | 2.95 ± 0.01[b] |
| C22:4n6 (g/100 g) | 1.16 ± 0.02[a] | 1.02 ± 0.01[b] | 0.99 ± 0.02[b] |
| SFA[b] (g/100 g) | 159.90 ± 1.94[b] | 165.31 ± 1.66[b] | 277.04 ± 4.30[a] |
| MUFA[c] (g/100 g) | 137.41 ± 1.24[c] | 148.54 ± 2.06[b] | 232.43 ± 0.01[a] |
| PUFA[d] (g/100 g) | 24.23 ± 0.25[b] | 22.32 ± 0.60[c] | 26.79 ± 0.06[a] |
| PUFA: SFA ratio | 0.15 ± 0.01[a] | 0.14 ± 0.01[b] | 0.10 ± 0.01[c] |
| n-3 PUFA[e] (g/100 g) | 2.59 ± 0.04[a] | 1.73 ± 0.03[c] | 2.04 ± 0.02[b] |
| n-6 PUFA[f] (g/100 g) | 21.63 ± 0.22[b] | 20.59 ± 0.60[b] | 24.75 ± 0.04[a] |
| n-6: n-3 PUFA ratio | 8.35 ± 0.06[b] | 11.91 ± 0.30[a] | 12.14 ± 0.11[a] |

[a]Within a row, means without a common superscript letter (a, b, c) are different ($P < 0.05$). SFA, saturated fatty acid; MUFA, monounsaturated fatty acid; PUFA, polyunsaturated fatty acid. NPH, non-pelleted native grass hay with 40% concentrate diet, PHLC, pelleted native grass hay with 40% concentrate diet; PHHC, pelleted native grass hay with 60% concentrate diet.
[b]SFA = C14:0 + C16:0 + C18:0 + C20:0 + C22:0 + C24:0.
[c]MUFA = C14:1 + C16:1 + C18:1n9c + C18:1n9t.
[d]PUFA = C18:2n6c + C18:2n6t + C18:3n3 + C20: 3n6 + C20:4n6.
[e]n-3 PUFA = C18:3n3.
[f]n-6 PUFA = C18:2n6c + C20:3n6 + C22:4n6.

from the three treatments. In addition, 1,343,882 valid reads were also obtained for the 18 samples from the three treatments, with the sequences of 70,573–78,336 for each sample, and these valid reads were clustered into 19,949 ASVs with the confidence threshold value of 80%. The Good's coverage of the NPH, PHLC, and PHHC treatments exceeded 0.99, indicating the sequencing depth perfectly meet the demand for the diversity analysis of the rumen microbiota (Table 6). Compared to the PHLC group, the ASV numbers in the PHHC group were significantly ($P < 0.05$) decreased, but no significant ($P > 0.05$) differences were found between the NPH and PHLC, NPH, and PHHC groups. The changes of the Chao1 value among the three groups were similar with the ASV numbers. There was also no significant ($P > 0.05$) difference on the Shannon index in the three groups.

The Venn diagram (Fig. 1A) showed 2,397, 2,409, and 1,834 unique ASVs in the NPH, PHLC, and PHHC groups, respectively, while all groups shared 945 ASVs. Additionally, the Venn diagram also revealed 509, 414, and 453 ASVs shared by the NPH and PHLC groups, NPH and PHHC groups, PHLC and PHHC groups, respectively. The PCoA plot of the ruminal microbiota in the NPH, PHLC, and PHHC groups was estimated by the Bray-Curtis distance metrics. The result indicated that 11.24% of the variation was captured by axis 1, while axis 2 represents 8.97% of the variation (Fig. 1B). Moreover, there was a markedly distinct separation, and the PERMANOVA showed significant difference ($P = 0.001$) among the three groups.

In the current study, 1,343,882 valid reads were obtained via 16S RNA sequencing, and the overall sequences were assigned into 20 phyla. Twenty phyla were allocated into the NPH group, 19 phyla were allocated into the PHLC group, and 18 phyla were allocated into the PHHC group (Fig. 1C). When examining the microbial composition

**TABLE 5** Effects of the diets on free amino acid profile of lamb[a]

| Item | NPH | PHLC | PHHC |
|---|---|---|---|
| Lysine (g/100 g) | 1.42 ± 0.03[ab] | 1.50 ± 0.02[a] | 1.35 ± 0.04[b] |
| Methionine (g/100 g) | 0.43 ± 0.01[ab] | 0.45 ± 0.01[a] | 0.40 ± 0.02[b] |
| Valine (g/100 g) | 0.82 ± 0.02[ab] | 0.85 ± 0.01[a] | 0.79 ± 0.01[b] |
| Isoleucine (g/100 g) | 0.80 ± 0.01[a] | 0.80 ± 0.01[a] | 0.73 ± 0.01[b] |
| Leucine (g/100 g) | 1.43 ± 0.01[a] | 1.43 ± 0.01[a] | 1.31 ± 0.02[b] |
| Phenylalanine (g/100 g) | 0.58 ± 0.01[ab] | 0.60 ± 0.01[a] | 0.56 ± 0.01[b] |
| Histidine (g/100 g) | 0.78 ± 0.01[b] | 0.81 ± 0.01[a] | 0.79 ± 0.01[ab] |
| Threonine (g/100 g) | 0.81 ± 0.02[ab] | 0.83 ± 0.01[a] | 0.77 ± 0.01[b] |
| Cystine (g/100 g) | 0.19 ± 0.01[a] | 0.17 ± 0.01[b] | 0.16 ± 0.01[c] |
| Alanine (g/100 g) | 1.01 ± 0.03[ab] | 1.04 ± 0.01[a] | 0.98 ± 0.01[b] |
| Aspartate (g/100 g) | 1.71 ± 0.02[a] | 1.75 ± 0.01[a] | 1.60 ± 0.03[b] |
| Glutamate (g/100 g) | 2.99 ± 0.01[a] | 2.98 ± 0.02[a] | 2.74 ± 0.04[b] |
| Arginine (g/100 g) | 1.97 ± 0.02[a] | 1.95 ± 0.03[a] | 1.82 ± 0.02[b] |
| Glycine (g/100 g) | 0.78 ± 0.02[a] | 0.79 ± 0.01[a] | 0.71 ± 0.02[b] |
| Serine (g/100 g) | 0.74 ± 0.01[a] | 0.74 ± 0.01[a] | 0.68 ± 0.01[b] |
| Tyrosine (g/100 g) | 0.56 ± 0.02[ab] | 0.59 ± 0.01[a] | 0.54 ± 0.01[b] |
| Proline (g/100 g) | 0.63 ± 0.01[a] | 0.63 ± 0.02[a] | 0.57 ± 0.02[b] |
| FAA[b] | 8.46 ± 0.09[a] | 8.50 ± 0.05[a] | 7.85 ± 0.11[b] |

[a]Within a row, means without a common superscript letter (a, b, c) are different ($P < 0.05$). FAA, flavor amino acids; EAA, essential amino acids; NEAA, non-essential amino acids; NPH, non-pelleted native grass hay with 40% concentrate diet, PHLC, pelleted native grass hay with 40% concentrate diet; PHHC, pelleted native grass hay with 60% concentrate diet.
[b]FAA = glutamate + aspartate + alanine + arginine + glycine.

among the diets at the phylum level, most of the sequencing belonged to *Bacteroidetes*, *Firmicutes*, and *Proteobacteria*. The *Bacteroidetes* was more dominant among the three diets with significant ($P < 0.05$) difference among the three group (Fig. 1D). The diets also significantly ($P < 0.05$) altered the relative abundance of *Proteobacteria* in the three diets, and the highest relative abundance was observed in the NPH group, followed by that in the PHLC and PHHC groups. At the genus level, a total of 20 genera were assigned to be predominant (the relative abundance higher than 1% in at least one group, Fig. 1E). In the present study, 31 genera were tested with significant ($P < 0.05$) difference among the NPH, PHLC, and PHHC groups (Fig. 1F), and four genera were characterized according to the relative abundance higher than 1% in at least one group, including *Rikenellaceae*_RC9_gut_group ($P < 0.01$), *Prevotellaceae*_UCG-003 ($P < 0.01$), *Erysipelotrichaceae*_UCG-004 ($P = 0.02$), and *Succinivibrio* ($P = 0.03$).

In this study, the LEfSe displayed the statistically different taxonomies in the ruminal microbiota of lambs fed with the NPH, PHLC, and PHHC diets (LDA >3 and $P < 0.05$;(Fig. 2A and B). The relative abundance higher than 1% in at least one group was characterized in detail in the current study, the genera *Succinivibrio* and *Erysipelotrichaceae*_UCG-004 were enriched in the NPH group, and the genus *Rikenellaceae*_RC9_gut_group was concentrated in the PHLC group.

**TABLE 6** Effect of the diets on the diversity indices of ruminal microbiota of lambs[a]

| Item | NPH | PHLC | PHHC | Total no. |
|---|---|---|---|---|
| No. of sequences | 86,012 ± 514[a] | 82,851 ± 984[b] | 84,677 ± 1,119[ab] | 1,521,250 |
| No. of valid sequences | 74,735 ± 695[a] | 73,999 ± 963[a] | 75,251 ± 839[a] | 1,343,882 |
| Observed ASV number | 1,016 ± 60[ab] | 1,109 ± 34[a] | 944 ± 36[b] | |
| Chao1 value | 1,075.14 ± 62.83[ab] | 1,171.23 ± 38.00[a] | 987.92 ± 39.99[b] | |
| Shannon index | 8.04 ± 0.20 | 8.28 ± 0.22 | 7.94 ± 0.20 | |
| Good's coverage | >0.99 | >0.99 | >0.99 | |

[a]Within a row, means without a common superscript letter (a, b, c) are different ($P < 0.05$). ASV, amplicon sequence variants. NPH, non-pelleted native grass hay with 40% concentrate diet, PHLC, pelleted native grass hay with 40% concentrate diet; PHHC, pelleted native grass hay with 60% concentrate diet.

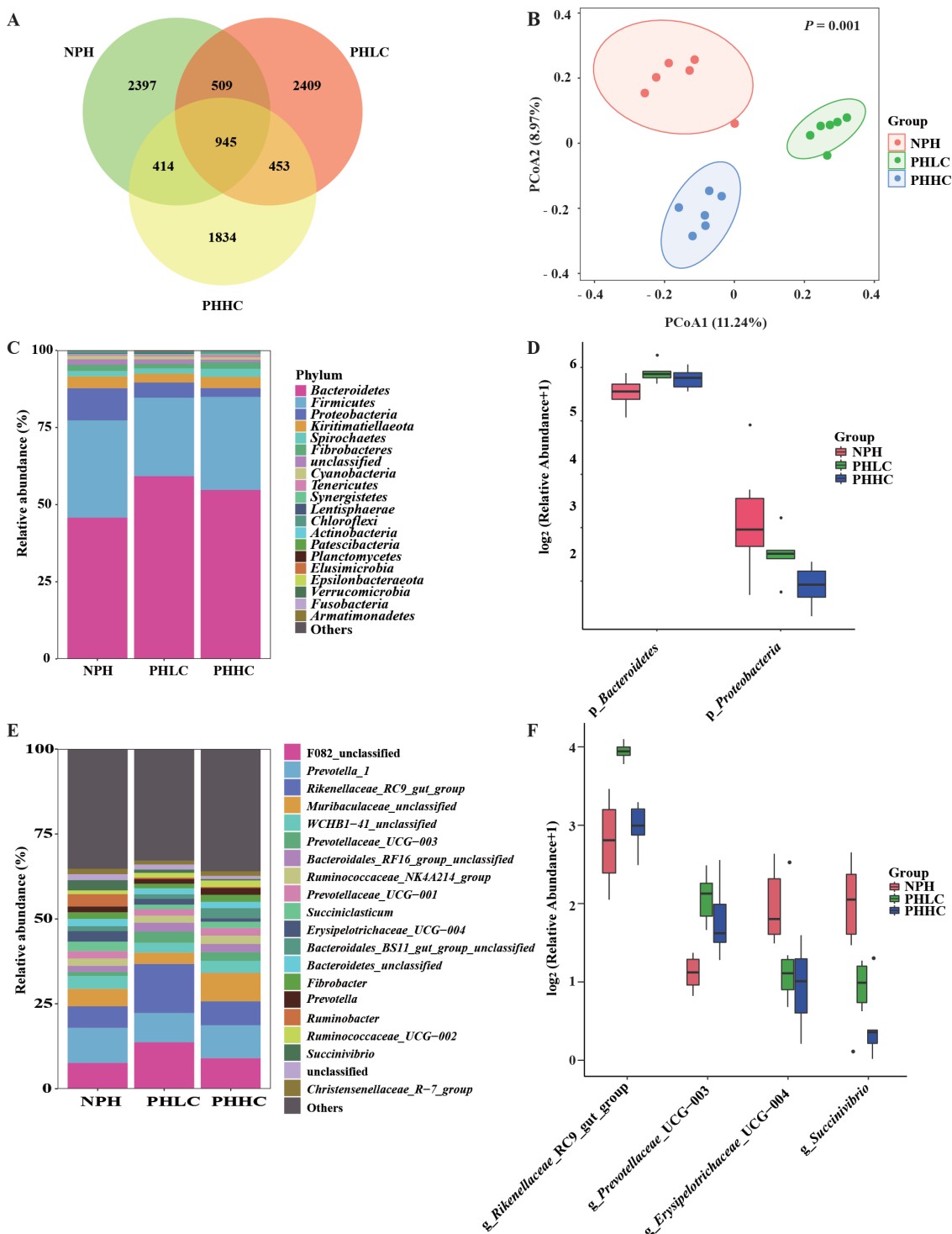

FIG 1 (A) Venn diagram representing the common and unique ASVs found at each diet. (B) Principal coordinate analysis of ruminal microbiota based on unweighted UniFrac distance. (C) The relative abundance (%) of bacterial phyla (1% at least in one group) of ruminal microbiota of lambs. (D) Extended error bar plot showing the bacteria at the phylum level that had significant differences among the NPH, PHLC, and PHHC groups. (E) The relative abundance (%) of bacterial genus (1% at least in one group) of ruminal microbiota of lambs. (F) Extended error bar plot showing the bacteria at the genus level that had significant differences among the NPH, PHLC, and PHHC groups. NPH, non-pelleted native grass hay with 40% concentrate diet; PHLC, pelleted native grass hay with 40% concentrate diet; PHHC, pelleted native grass hay with 60% concentrate diet.

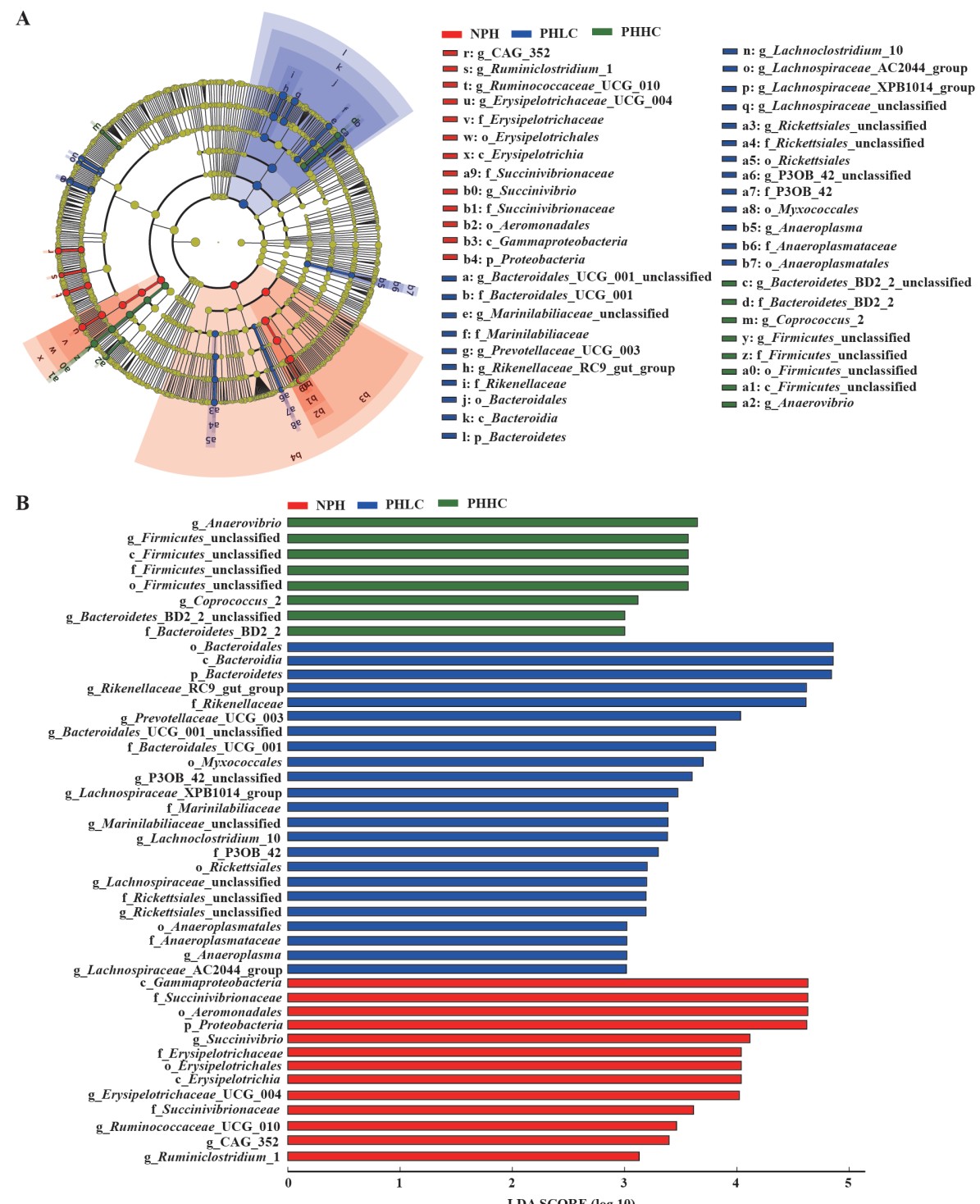

**FIG 2** Linear discrimination analysis coupled with effect size analysis of the ruminal microbiota of lamb in the NPH, PHLC, and PHHC groups. (A) Cladogram showing microbial species with significant differences among the three treatments. Red, green, and blue represent different groups. Species classification at the phylum, class, order, family, and genus levels are displayed from inner to outer layers. The red, green, and blue nodes represent microbial species in the phylogenetic tree, which play important roles in the NPH, PHLC, and PHHC groups, respectively. Yellow nodes represent no significant difference between species. (B) Significantly different species with an LDA score greater than the estimated value (default score = 3 and *P* < 0.01). The length of the histogram represents the LDA score of different species in the three groups. NPH, non-pelleted native grass hay with 40% concentrate diet; PHLC, pelleted native grass hay with 40% concentrate diet; PHHC, pelleted native grass hay with 60% concentrate diet.

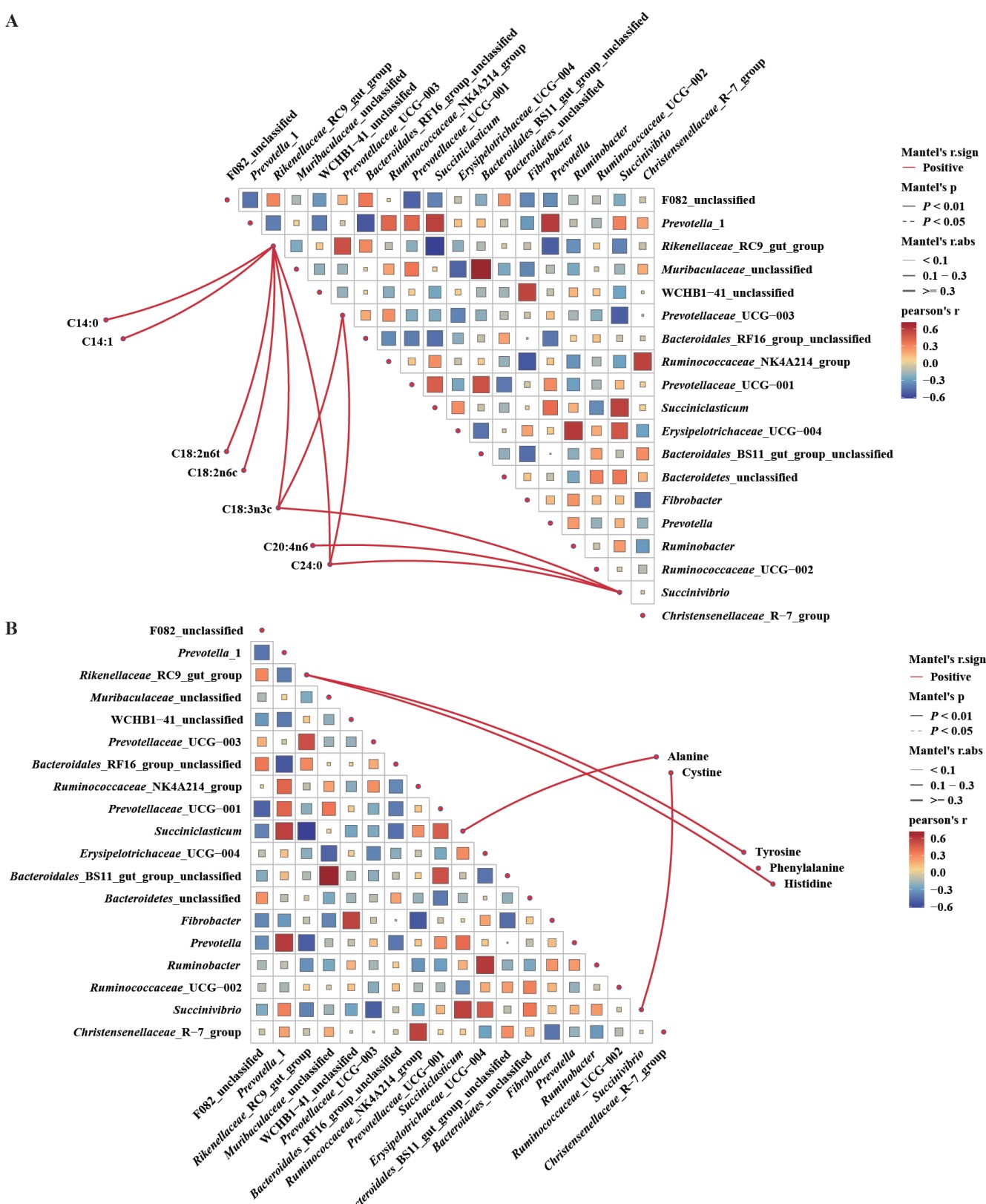

**FIG 3** Mantel analysis between the composition of rumen microbiota and fatty acid or amino acid profiles. (A) Mantel analysis between the composition of rumen microbiota and fatty acid profiles. (B) Mantel analysis between the composition of rumen microbiota and fatty acid profiles. Red, positive correlation; blue, negative correlation. The square chart indicates the magnitude of the correlation coefficients, and the shade of color indicates the significance of the correlation coefficients.

## Mantel analysis between predominant rumen bacteria and fatty acid profile or amino acid profile

To delve the fatty acid or amino acid profiles for plastisphere microbial variation, the Mantel test between the composition of rumen microbiota and fatty acid or amino acid profiles was performed (Fig. 3). The *Rikenellaceae*_RC9_gut_group was significantly correlated with most fatty acid profiles (Fig. 3A), including C18:3n3 ($r = 0.466$, $P < 0.01$) and C24:0 ($r = 0.477$, $P < 0.01$). The genus *Prevotellaceae*_UCG-003 was markedly associated with the C18:3n3c ($r = 0.380$, $P < 0.01$) and C24:0 ($r = 0.387$, $P < 0.01$). Lastly, the genus *Succinivibrio* was also markedly associated with the C18:3n3c ($r = 0.320$, $P < 0.01$) and C24:0 ($r = 0.351$, $P < 0.01$). Mantel correlations also examined the relationships between the amino acid profiles of the longissimus dorsi muscle and the rumen microbiota (Fig. 3B). Similarly, the genus *Rikenellaceae*_RC9_gut_group was significantly correlated with most amino acid profiles, including histidine ($r = 0.360$, $P < 0.01$), tyrosine ($r = 0.334$, $P < 0.01$), and alanine ($r = 0.299$, $P < 0.01$). Additionally, the genera *Succinivibrio* and *Succiniclasticum* also markedly associated with some amino acid profiles, such as cystine ($r = 0.377$, $P < 0.01$) and alanine ($r = 0.463$, $P < 0.01$).

## DISCUSSION

Rumen microbiota is increasingly recognized not only for their crucial role in modulating the growth and health of the host but also in improving the fatty acid and amino acid profiles deposition in ruminant products (30–32). Diet has emerged as a driving factor in alternating the compositions and functions of the rumen microbiota (33, 34). In the present study, a combination of multiple physicochemical analyses and high-throughput sequencing was employed to investigate the influences of NPH, PHLC, and PHHC diets on the animal performance, meat quality, fatty acid compositions, amino acid profiles, and rumen microbiota. This is the first study that integrates multiple physicochemical analyses and rumen microbiota in response to native grass-based diet influencing the edibility of lamb, providing a tentative model for future research in this area.

Pelleted or concentrate supplementation diets have been proved as an effective strategy to enhance animal growth (11, 35). Interestingly, the ADG and BW gain increased in the PHLC group than that in the NPH and PHLC groups, which could be explained by the PHLC diet containing lower fiber and higher energy as compared with the NPH and PHLC diets (36). On the other hand, no significant difference was observed between the NPH and PHLC groups, which is similar with a previous report that found that hay versus pelleted hay had no effects on the animal performance (37). In the present study, there were no differences on the DMI, which is in accordance with Trabi who also found that the feed pelleted low-grain or non-pelleted high-grain total mixed ration had no effects on the DMI (36). The sufficient fiber and energy intakes provided by the three diets might be the main reason. Carcass weight is an important parameter for evaluating the utilization of diet nutrient. In this research, although the significantly higher ADG was observed in the PHHC group, there were no significant differences in carcass weight, dressing percentages, or backfat thickness. This is consistent with the previous findings that concentrate increase from 40% to 60% did not change the carcass weight and dressing percentage (38, 39).

In recent years, the meat products with less fat content are considered healthier by the consumers (40). The markedly higher intramuscular fat and lower protein contents were found in the PHHC group than that in the NPH and PHLC groups, which could be attributed to the higher fatty acid contents in the concentrate, which had positive effects on deposition (40, 41). The initial ($pH_{45\ min}$) and ultimate ($pH_{24\ h}$) pH values decreased from 6.56 to 5.40, within the acceptable range. The $pH_{45\ min}$ after slaughter was influenced by the diet, and the $pH_{24\ h}$ was similar with previously reported values (42). The ultimate pH is related to the degradation of glycogen during the post slaughter periods and is not substantially affected by diet (39). Although the meat color is only weakly correlated with the meat flavor, it could directly affect the consumers' choice. There was a significant difference in $L^*$ and $b^*$ between the NPH and PHLC groups, which

is contrast with the previous report that the feed physical form had no significant effect on meat color (43).

The nutritional value and oxidative stability were determined by the fatty acid and amino acid profiles of meat. Therefore, the author also investigated the effects of diets on the compositions of fatty acids and amino acids of lamb. In the current study, the authors found that most of the fatty acid profile was altered by the diet. All of the groups showed the prevalence of C16:0, C18:1n9c, and C18:0 contents in lamb meat, which is in agreement with that of other reports (44). In the present study, the PHHC diet increased the C16:0, C18:1n9c, and C18:0 contents in lamb meat, which could be attributed to the higher intake of concentrate. The ratio of PUFA to SFA is important because the higher intake of SFA has been reported to increase the risk of heart disease and type 2 diabetes, whereas the PUFA (mainly C20:4n6) could possess numerous benefits (45, 46). The concentrate diet might be the main reason for the higher SFA content and lower PUFA to SFA ratio in the PHHC group, which could be explained by the metabolic pathway of fatty acid biosynthesis in sheep, which can be enhanced in the concentrate diet (12, 47). Furthermore, the fatty acid contents of meat largely depend on the dietary fatty acid sources and ruminal bacterial (47, 48). In this study, there were no significant differences in most of the amino acids between the NPH and PHLC diets, indicating that the physical form had no significant effect on the compositions of amino acids (43). Nevertheless, the PHHC diet significantly influenced the compositions of amino acids compared to that in the NPH and PHLC groups. The higher intake of amino acid in the concentrate might be the main reason. Interestingly, the flavor amino acid, including glutamate, aspartate, alanine, and arginine, was markedly decreased in the PHHC group than that in the NPH and PHLC groups. These findings are in line with the previous reports (16). To date, the exact mechanism that modulates compositions of amino acid is still unclear.

These results suggest that diet could influence the rumen bacterial community composition in lambs. Higher ASV number and Chao1 index were observed in the NPH and PHLC groups compared to the PHHC group. These results were similar to Liu et al. who found grass diet-fed lambs had higher bacterial diversity and richness than concentrate diet-fed animals (49). No significant differences were observed in the Shannon index among the three groups. The changes in the rumen bacterial compositions were also explored; the present results suggest that the NPH, PHLC, and PHHC groups have its distinct microbiota, as reflected by the clustering of the samples by diet group using PCoA. Macroscopically, the different diets drove a separation in the bacterial community, the distinguishable changes among the three groups, following the reports that noticeable separation of the microbial structure was observed among non-pelleted low-grain, pelleted low-grain, and high-grain diets (36, 50), which could be attributed to the growth of microorganisms under various pH conditions (51).

This study revealed some differences in the rumen microbiota among the three diets, particularly the phyla *Bacteroidetes* and *Proteobacteria*, including the genus *Prevotellaceae*_UCG-003, *Succinivibrio*, and *Erysipelotrichaceae*_UCG-004. The genus *Prevotellaceae*_UCG-003 can reduce the nitrogen losses and produce acetic and succinic acids as the fermentation end-products of glucose (49). The genus *Prevotellaceae*_UCG-003 decreased in the NPH group compared to that in the PHLC and PHHC groups. These results were in accordance with the previous report that the pelleted diet could increase the abundance of *Prevotellaceae* (52). The genus *Erysipelotrichaceae* is tightly associated with gut health (53). In this study, the abundance of *Erysipelotrichaceae* decreased in the pelleted diet, which is in agreement with a previous report that found that pelleted diet decreased the abundance of *Erysipelotrichaceae* and might affect gut development (52). The genus *Succinivibrio*, two short-chain fatty acid-producing bacteria (54, 55), was significantly decreased in the PHHC group. These results were similar to prior reports that found that grass diet increased the abundance of *Succinivibrio* (56).

Ruminant meat is characterized by having considerable percentages of fatty acids and amino acids, and these profiles are subjected to a process of biohydrogenation

conducted by ruminal microbiota (57), indicating that the fatty acid and amino acid profiles were activated by rumen microbiota. The *Rikenellaceae*_RC9_gut_group was the butyrate-producing bacteria and could increase the AMPK activity to regulate the lipid deposition traits by changing the production of volatile fatty acid (VFAs) (58). Furthermore, the higher and lower abundances of *Prevotellaceae*_UCG-003 and *Succinivibrio* were found in the PHHC treatment compared to that in the NPH treatment. The previous reports also indicated that the genera *Prevotellaceae*_UCG-003 and *Succinivibrio* are efficiently metabolize into fatty acid synthesis (6, 59). Therefore, the significant differences were observed in the fatty acid profiles among these treatments. Therefore, these genera were also correlated with some of the fatty acids of meat. Furthermore, the amino acids were also associated with the rumen microbiota, such as *Rikenellaceae*_RC9_gut_group, *Succinivibrio*, and *Succiniclasticum*, which follows previous research that found *Rikenellaceae*_RC9_gut_group and *Succinivibrio* are associated with amino acid (60, 61).

## Conclusion

In summary, this study involved a combination of physicochemical analyses and 16S RNA sequencing analyses; the associations between the specific bacterial genera and fatty acid and amino acid profiles were significantly influenced by the diet. These results could provide a better understanding of meat quality, fatty acid, amino acid profiles, and microbial functions that contribute to the development of modern lamb husbandry strategies. Furthermore, the causes and mechanisms driving the interactions among ruminal microbiota, serum metabolism, and meat quality merit further investigation.

## ACKNOWLEDGMENTS

This work was supported by the Development and Utilization of Forage for Mutton in Inner Mongolia (s20021) and Technology Project of Inner Mongolia (2020GG0032), China.

T.L.: conceptualization and writing—original draft. Z.B.: conceptualization and investigation. K.X.: investigation. Y.J.: supervision. S.D.: conceptualization, formal analysis, and writing—review and editing.

We declare that we have no financial and personal relationships with other people or organizations that can inappropriately influence our work, and there is no professional or other personal interest of any nature or kind in any product, service, and/or company that could be construed as influencing the content of this paper.

## AUTHOR AFFILIATIONS

[1]College of Agriculture, Inner Mongolia University of Nationalities, Tongliao, China
[2]Guangdong Laboratory of Lingnan Modern Agriculture, Agriculture Genomics Institute, Chinese Academy of Agricultural Sciences, Shenzhen, China
[3]Genome Analysis Laboratory of the Ministry of Agriculture and Rural Affairs, Agriculture Genomics Institute, Chinese Academy of Agricultural Sciences, Shenzhen, China
[4]Forest and Grassland Protection and Development Center, Chifeng, China
[5]Key Laboratory of Forage Cultivation, Processing and High Efficient Utilization, Ministry of Agriculture, College of Grassland, Resources and Environment, Inner Mongolia Agricultural University, Hohhot, Inner Mongolia, China

## AUTHOR ORCIDs

Shuai Du http://orcid.org/0000-0002-2035-0927

## FUNDING

| Funder | Grant(s) | Author(s) |
| --- | --- | --- |
| Development and Utilization of Forage for Mutton in Inner Mongolia | s20021 | Tingyu Liu |
| Technology Project of Inner Mongolia | 2020GG0032 | Yushan Jia |

## AUTHOR CONTRIBUTIONS

Tingyu Liu, Conceptualization, Writing – original draft | Zhenkun Bu, Conceptualization, Investigation | Kaifeng Xiang, Investigation | Yushan Jia, Supervision | Shuai Du, Conceptualization, Formal analysis, Writing – review and editing

## DATA AVAILABILITY

The raw data sequencing files and associated metadata of these samples have been uploaded into the NCBI's Sequence Read Archive with the accession number PRJNA1025899.

## ETHICS APPROVAL

The animal study was reviewed and approved by the College of Grassland, Resources and Environment, Inner Mongolia Agricultural University. All experiments were performed according to the Regulation on the Administration of Laboratory Animals (The State Science and Technology Commission of China, 2017).

## ADDITIONAL FILES

The following material is available online.

### Open Peer Review

**PEER REVIEW HISTORY (review-history.pdf).** An accounting of the reviewer comments and feedback.

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
