## [Reviewer comments · Microbiology Spectrum]

Microbiology Spectrum

Effects of non-pelleted or pelleted low native grass, pelleted high native grass diets on meat quality by regulating the rumen microbiota in lambs

Tingyu Liu, Zhenkun Bu, Kaifeng Xiang, Yushan Jia, and Shuai Du

Corresponding Author(s): Shuai Du, Inner Mongolia Agricultural University

Review Timeline:

Submission Date:	October 23, 2023
Editorial Decision:	December 6, 2023
Revision Received:	January 22, 2024
Accepted:	January 30, 2024

Editor: Jing Han

Reviewer(s): The reviewers have opted to remain anonymous.

Transaction Report:

DOI: <https://doi.org/10.1128/spectrum.03758-23>

Re: Spectrum03758-23 (Effects of non-pelleted or pelleted low native grass, pelleted high native grass diets on meat quality by regulating the rumen microbiota in lambs)

Dear Dr. Shuai Du:

Thank you for the privilege of reviewing your work. Below you will find my comments, instructions from the Spectrum editorial office, and the reviewer comments.

Revision Guidelines

Sincerely,
Jing Han
Editor
Microbiology Spectrum

Reviewer #1 (Comments for the Author):

The study evaluated the different diet on meat quality in lambs and used suitable omics. The results is quit value. It is a interesting and value study to publish. However, there are some point preventing this paper from publishment.

Major comments:

1. L41-42: only bacterial composition (16S rRNA) was used and only correlation analysis was conducted. It is difficult to

conclude that the bacterial regulated the meat profiles. Please revise all.

2. L 59-71: The introduction is missing about the pelleted and no-pelleted diet, or native grass and unnative grass. We difficultly to understand why you do this study.

Minor comments:

1. L44 Lambs is also a important key word.
2. L169 80% is quite low to identify ASV. How to consider it?
3. L164 did you do the data filtering or data normalization? Please added it.
4. L190-192: Please add the model to calculate the data. What is the fixed factor?
5. L643: what is the native grass. Only one grass or the mixed grasses?
6. L643: many 0 in table 1. Please revise it.
7. L643: The ME is analyzed or calculated by format?
8. L644: (non-pelleted native grass hay with 40% concentrate diet). It means that the native grass was not pelleted and was fed to lambs. The grass and concentrate were fed to lambs individually?
9. 661: a, b, c is not need to mark when the P is less than 0.05.
10. 713: Please delete EAA and NEAA. There is no reference to define the EAA of lambs.
11. In figure 1B, how to obtain the P value?
12. A lot of information in Figure 3. Please simplify it and stress the important results. The title is wrong because the Mantel analysis was used and the correlation analysis is not important in this figure.

Reviewer #2 (Public repository details (Required)):

The raw 16S data has been deposited into the NCBI Sequence Read Archive database (Accession Number: PRJNA1025899).

Reviewer #2 (Comments for the Author):

This study determined the meat quality, fatty acid contents, amino acid profiles and rumen microbiota of lamb fed with non-pelleted or pelleted low native grass, pelleted high native grass diets by using 16S amplicon sequencing methods. It was found that the diets influence the meat quality, fatty acid and amino acid profiles by regulating the rumen microbiota. The presented results not only provide the effects of diet on the rumen microbiota but highlight the importance of the rumen microbiota on the meat quality, fatty acid and amino acid profiles. The article should be of interest for the readers but before publication, a depth revision should be addressed. The authors should pay much more attention in preparing the manuscript.

Specific suggestions are as follows:

The format of the manuscript should be revised based on the requirements.

Table 1. The chemical compositions of the diet should be added.

Abstract

1. The abstract provides information about the impact of different diets on lamb quality but lacks various details, such as the experimental design.
2. I believe the introduction does not explicitly state the hypothesis of this study. Such as how different diets reshape the microbial community to participate in the construction of the fatty acid profile.

Materials and methods

1. Please provide the nutritional composition data from Table 1. Why are all the nutritional components zero?
2. How were the rumen samples collected?
3. How many samples were used for DNA extraction and LC-MS? Please specify in the text.
4. Please provide specific parameters for gas chromatography in the determination of amino acids and fatty acids.
5. How the samples used for pH24h measurement were preserved? Similarly, how were the samples for amino acids and fatty acids stored and thawed before the analysis?

Discussion

1. Why the grass diet might be the main reason resulting in higher SFA content and lower PUFA to SFA ratio in the PHHC group?
2. You mentioned the association between Rikenellaceae_RC9_gut_group and the differences in fatty acid profiles in this study. However, why do PHHC and NPH groups show differences in most fatty acids while the relative abundance of Rikenellaceae_RC9_gut_group isn't notably different, as depicted in Figure 1F?

The study evaluated the different diet on meat quality in lambs and used suitable omics.

The results is quit value. It is a interesting and value study to publish. However, there are some point preventing this paper from publishment.

Major comments:

1. L41-42: only bacterial composition (16S rRNA) was used and only correlation analysis was conducted. It is difficult to conclude that the bacterial regulated the meat profiles. Please revise all.
2. L 59-71: The introduction is missing about the pelleted and no-pelleted diet, or native grass and unnative grass. We difficultly to understand why you do this study.

Minor comments:

1. L44 Lambs is also a important key word.
2. L169 80% is quite low to identify ASV. How to consider it?
3. L164 did you do the data filtering or data normalization? Please added it.
4. L190-192: Please add the model to calculate the data. What is the fixed factor?
5. L643: what is the native grass. Only one grass or the mixed grasses?
6. L643: many 0 in table 1. Please revise it.
7. L643: The ME is analyzed or calculated by format?
8. L644: (non-pelleted native grass hay with 40% concentrate diet). It means that the native grass was not pelleted and was fed to lambs. The grass and concentrate were fed to lambs individually?
9. 661: a, b, c is not need to mark when the P is less than 0.05.

10. 713: Please delete EAA and NEAA. There is no reference to define the EAA of lambs.

11. In figure 1B, how to obtain the P value?

12. A lot of information in Figure 3. Please simplify it and stress the important results.

The title is wrong because the Mantel analysis was used and the correlation analysis is not important in this figure.

Dear Editor and Reviewers

Thank you very much for evaluating our paper.

Thank you for your letter and for the reviewers' comments concerning our manuscript entitled "Effects of non-pelleted or pelleted low native grass, pelleted high native grass diets on meat quality by regulating the rumen microbiota in lambs" (No. Spectrum 03758-23). We will be happy to edit the text further, based on helpful comments from editor and reviewers. We appreciate the editor very much for their positive and constructive comments and suggestions. According to the comments, this revised manuscript was checked by native speakers of English for editing English grammar. The comments and suggestions are not only helpful for us to revise and improve our manuscript, but also benefit our further research. We hope that our paper much better quality than before.

Best regards,

Corresponding author:

Dr. Shuai Du

E-mail: dushai_nm@sina.com

Reviewer #1 (Comments for the Author):

Major comments:

1. L48-54: only bacterial composition (16S rRNA) was used and only correlation analysis was conducted. It is difficult to conclude that the bacterial regulated the meat profiles. Please revise all.

Response: Yes, we have revised in the manuscript.

「The correlation analysis of the association of microbiome with the meat quality provides us with comprehensive understanding of the composition and function of the rumen microbial community, and these findings will contribute to the direction of future research in lamb.」

2. L 104-140: The introduction is missing about the pelleted and no-pelleted diet, or native grass and unnative grass. We difficultly to understand why you do this study.

Response: Yes, we have revised clearly as:

「Traditionally, the lambs were thinner and weaker (even leading to death) when fed with native grass hay in winter and spring (Du et al., 2020). The native grass pellets or non-pelleted native grass hay supplements with concentrate diets have become common by increasing the dry matter intake (DMI) and the average daily gain (ADG) in sheep production systems (Du et al., 2022). Previously published research found that the bacteria play an important role in most of the feed biopolymer degradation and fermentation, which indicated that the bacteria are key players for animal performance (Xue et al., 2020). Nevertheless, the rumen microbiota is strongly influenced by individual genetics, animal age, feed type, and feeding system, especially is directly linked to the diet (Arshad et al., 2021). On the one hand, the low-forage diets could affect the composition of the rumen microbiota (Plaizier et al., 2012), and the high-forage diets are beneficial for the genus Firmicutes (Khafipour et al., 2009). On the other hand, previous reports also found that the pellet diets could reduce the bacterial richness in the rumen (McClements and Xiao, 2012), whereas others found that the pelleted-hay diets had a greater increase in bacterial richness

(Ishaq et al., 2019). Compared to the native grass pellet supplements with concentrate diets, the pelleted native grass hay could increase the animal performance and the bacterial abundance in the rumen (Du et al., 2023). The alternation of rumen microbiota in turn influences the meat quality *via* the intramuscular fatty acid and amino acid metabolism, such as *Prevotella*, *Clostridiales* and *Ruminococcaceae* are likely related to lipid and protein metabolism (Huws et al., 2011; Vasta et al., 2010; Wang et al., 2019). However, to our knowledge, little information is available on the effect of the non-pelleted or pelleted native grass diets on the meat quality, fatty acid and amino acid profiles, composition and function of the rumen microbiota.」

Minor comments:

1. L55 Lambs is also a important key word.

Response: Yes, we have added lamb as a key word in the abstract.

2. L268 80% is quite low to identify ASV. How to consider it?

Response: There was a mistake. We have revised in the manuscript.

3. L259 did you do the data filtering or data normalization? Please added it.

Response: Yes, we have revised.

「The low-quality reads (quality scores lower than 20) and short reads (lower than 100 bp) were trimmed by using the sliding-window algorithm method in fqtrim (v 0.94). Quality filtering was performed to obtain high-quality clean tags according to fqtrim. Chimeric sequences were filtered using Vsearch software (v2.3.4).」

4. L289-300: Please add the model to calculate the data. What is the fixed factor?

Response: Yes, we have revised in the manuscript.

「All data were calculated from measurements collected throughout the study and analyzed as repeated measures using the PROC MIXED procedure of SAS (SAS Inst., Inc., Cary, NC). Significant differences among groups were analyzed by a one-way

analysis of variance according to the statistical model, $Y = \mu + \alpha + \varepsilon$, where Y = observation, μ = overall mean, α = diet effect, and ε = error, and Duncan's tests, with a $p < 0.05$ as statistical significance using SAS ver. 9.0 and the diet as fixed factors.」

5. L643: what is the native grass. Only one grass or the mixed grasses?

Response: The native grass was mixed grass and harvested from the typical steppe in Xilinhot, Inner Mongolian Plateau, the native grass including *Stipa gigantea* L., *Leymus chinensis* (Trin.) Tzvel., *Lespedeza davurica* (Laxm.) Schindl, *Allium mongolicum* Regel, *Thalictrum petaloideum* Linn., *Bupleurum chinensis* DC., *Serratula centauroides* Linn., *Caragana microphylla* Lam and others.

6. Many 0 in table 1. Please revise it.

Response: Yes, we have revised in the manuscript.

7. L171: The ME is analyzed or calculated by format?

Response: Yes, we have revised clearly as:

「The metabolizable energy was calculated according to the previously published method of the following formula: $ME = GE - FE - UE - Eg$ (Bu et al., 2021).」

8. L160-167: (non-pelleted native grass hay with 40% concentrate diet). It means that the native grass was not pelleted and was fed to lambs. The grass and concentrate were fed to lambs individually?

Response: Yes, we have revised clearly as:

「The lambs were received three diets: (1) non-pelleted native grass hay with 40% concentrate diet (NPH), the native grass and concentrate were fed individually, (2) pelleted native grass hay with 40% concentrate diet (PHLC), (3) pelleted native grass hay with 60% concentrate diet (PHHC).」

9. Table: a, b, c is not need to mark when the P is less than 0.05.

Response: Yes, we have revised throughout the manuscript.

「^{a, b} Within a row, means without a common superscript are different ($p < 0.05$).」

10. Table 5: Please delete EAA and NEAA. There is no reference to define the EAA of lambs.

Response: Yes, we have deleted.

11. In figure 1B, how to obtain the P value?

Response: Yes, we have revised in the manuscript.

「The permutational multivariate analysis of variance (PERMANOVA) test was applied to analyze the significant difference among these treatments with R (version 2.5.4) software (Chambers and Hastie, 1992).」

12. A lot of information in Figure 3. Please simplify it and stress the important results. The title is wrong because the Mantel analysis was used and the correlation analysis is not important in this figure.

Response: Yes, we have revised in the manuscript.

「To delve the fatty acid or amino acid profiles for plastisphere microbial variation, the Mantel test between the composition of rumen microbiota and fatty acid or amino acid profiles were performed (Fig. 3). The *Rikenellaceae_RC9_gut_group* was significantly correlated with most fatty acid profiles (Fig. 3A), including C18:3n3 ($r = 0.466$, $p < 0.01$), C24:0 ($r = 0.477$, $p < 0.01$). The genus *Prevotellaceae_UCG-003* was markedly associated with the C18:3n3c ($r = 0.380$, $p < 0.01$) and C24:0 ($r = 0.387$, $p < 0.01$). At last, the genus *Succinivibrio* also markedly associated with the C18:3n3c ($r = 0.320$, $p < 0.01$) and C24:0 ($r = 0.351$, $p < 0.01$). Mantel correlations were also examined the relationships between the amino acid profiles of the *longissimus dorsi* muscle and the rumen microbiota (Fig. 3B). Similarly, the genus *Rikenellaceae_RC9_gut_group* was significantly correlated with most amino acid

profiles, including histidine ($r = 0.360$, $p < 0.01$), tyrosine ($r = 0.334$, $p < 0.01$) and alanine ($r = 0.299$, $p < 0.01$). Additionally, the genera *Succinivibrio* and *Succiniclasticum* also markedly associated with some amino acid profiles, such as cystine ($r = 0.377$, $p < 0.01$) and alanine ($r = 0.463$, $p < 0.01$).」 & 「Mantel analysis between the composition of rumen microbiota and fatty acid or amino acid profiles. (A) Mantel analysis between the composition of rumen microbiota and fatty acid profiles; (B) Mantel analysis between the composition of rumen microbiota and fatty acid profiles. Red, positive correlation; blue, negative correlation. The square chart indicates the magnitude of the correlation coefficients, and the shade of color indicates the significance of the correlation coefficients.」

Reviewer #2 (Public repository details (Required)):

The raw 16S data has been deposited into the NCBI Sequence Read Archive database (Accession Number: PRJNA1025899).

Response: Yes, we have revised in the manuscript.

Reviewer #2 (Comments for the Author):

Table 1. The chemical compositions of the diet should be added.

Response: Yes, we have revised in the manuscript.

Abstract

1. The abstract provides information about the impact of different diets on lamb quality but lacks various details, such as the experimental design.

Response: Yes, we have revised in the manuscript.

「Diet modulates the rumen microbiota, which in turn can impact the animal performance. The rumen microbiota is increasingly recognized for its crucial role in regulating the growth and meat quality of the host. Nevertheless, the mechanism by which the rumen microbiome influences the fatty acid and amino acid profiles of

lambs in the grass feeding system remains unclear.」

2. L104-140 I believe the introduction does not explicitly state the hypothesis of this study. Such as how different diets reshape the microbial community to participate in the construction of the fatty acid profile.

Response: Yes, we have revised in the manuscript.

「Traditionally, the lambs were thinner and weaker (even leading to death) when fed with native grass hay in winter and spring (Du et al., 2020). The native grass pellets or non-pelleted native grass hay supplements with concentrate diets have become common by increasing the dry matter intake (DMI) and the average daily gain (ADG) in sheep production systems (Du et al., 2022). Previously published research found that the bacteria play an important role in most of the feed biopolymer degradation and fermentation, which indicated that the bacteria are key players for animal performance (Xue et al., 2020). Nevertheless, the rumen microbiota is strongly influenced by individual genetics, animal age, feed type, and feeding system, especially is directly linked to the diet (Arshad et al., 2021). On the one hand, the low-forage diets could affect the composition of the rumen microbiota (Plaizier et al., 2012), and the high-forage diets are beneficial for the genus Firmicutes (Khafipour et al., 2009). On the other hand, previous reports also found that the pellet diets could reduce the bacterial richness in the rumen (McClements and Xiao, 2012), whereas others found that the pelleted-hay diets had a greater increase in bacterial richness (Ishaq et al., 2019). Compared to the native grass pellet supplements with concentrate diets, the pelleted native grass hay could increase the animal performance and the bacterial abundance in the rumen (Du et al., 2023). The alternation of rumen microbiota in turn influences the meat quality *via* the intramuscular fatty acid and amino acid metabolism, such as *Prevotella*, *Clostridiales* and *Ruminococcaceae* are likely related to lipid and protein metabolism (Huws et al., 2011; Vasta et al., 2010; Wang et al., 2018). However, to our knowledge, little information is available on the effect of the non-pelleted or pelleted native grass diets on the meat quality, fatty acid and amino acid profiles, composition and function of the rumen microbiota. 」

Materials and methods

1. Please provide the nutritional composition data from Table 1. Why are all the nutritional components zero?

Response: Yes, we have added the nutritional composition in the Table 1.

2. L 240-241 How were the rumen samples collected? How many samples were used for DNA extraction? Please specify in the text.

Response: Yes, we have revised in the manuscript.

「One lamb from each cage was randomly selected and a total of 18 lambs were sampled for analyzing the rumen microbiome.」

3. L219-236 Please provide specific parameters for gas chromatography in the determination of amino acids and fatty acids.

Response: Yes, we have revised clearly as:

「The fatty acids profiles were measured according to the published method with a gas chromatography–mass spectrometer 7890B (Agilent, California, United States) (Bu et al., 2021). The samples were melted in a steam bath or oven at 10 °C above the melting point. If the melted fat was cloudy, it was filtered through filter paper. Methyl esters of fatty acids were prepared from 400 to 500 mg fat. Before infrared analysis, excess impurities were removed with a suitable cleanup procedure. Then the undiluted fatty acid methyl esters were weighed to the nearest 0.1 mg into a 25-mL volumetric flask. A cell was filled with CS₂ solution, and a matching cell was filled with a test sample. Finally, the test sample or calibration solution was scanned in the same range as that of the reference. The amino acid profiles were measured according to the published method with an automatic amino acid analyzer (L-8800, Hitachi Ltd, Tokyo, Japan) (Hao et al., 2020). Briefly, the samples were added to 15 mL of 6 mol/L HCl with three to four drops of phenol. After hydrolyzation at 110 ± 1 °C for 22 h under nitrogen, the samples were filtrated and 1 mL supernatant was evaporated in a

vacuum drying oven at 50 °C, then redissolved in 1 mL of saline sodium citrate.」

4. L202-204 How the samples used for pH24h measurement were preserved? Similarly, how were the samples for amino acids and fatty acids stored and thawed before the analysis?

Response: Yes, we have revised in the manuscript.

「Subsequently, about 500 g of the *longissimus dorsi* muscle were taken from the left side of the carcass, placed in self-sealed bags, and then stored at 4 °C for analysis of meat quality and meat nutritional value.」

Discussion

1. L482- 489 Why the grass diet might be the main reason resulting in higher SFA content and lower PUFA to SFA ratio in the PHHC group?

Response: There was a mistake. We have revised in the manuscript.

「The concentrate diet might be the main reason resulting in higher SFA content and lower PUFA to SFA ratio in the PHHC group, which could be explained by the metabolic pathway of fatty acid biosynthesis in sheep can be enhanced in the concentrate diet (Jin et al., 2021, Du et al., 2023). Furthermore, the fatty acid contents of meat largely depend on the dietary fatty acid sources and ruminal bacterial (Jin et al., 2021; Sinclair et al., 2005). 」

2. L533L548 You mentioned the association between Rikenellaceae_RC9_gut_group and the differences in fatty acid profiles in this study. However, why do PHHC and NPH groups show differences in most fatty acids while the relative abundance of Rikenellaceae_RC9_gut_group isn't notably different, as depicted in Figure 1F?

Response: Yes, we have revised clearly as:

「The *Rikenellaceae_RC9_gut_group* was the butyrate-producing bacteria and could increase the AMPK activity to regulate the lipid deposition traits by changes the production of VFAs (Cheng et al., 2022). Furthermore, the higher and lower abundance of *Prevotellaceae_UCG-003* and *Succinivibrio* were found in the PHHC

treatment compared to that in the NPH treatment. The previous reports also indicated that the genus *Prevotellaceae*_UCG-003 and *Succinivibrio* are efficiently metabolize into fatty acid synthesizing (Lean et al., 2013; Du et al., 2022). Therefore, the significant differences were observed on the fatty acid profiles among these treatments. ┘

Re: Spectrum03758-23R1 (Effects of non-pelleted or pelleted low native grass, pelleted high native grass diets on meat quality by regulating the rumen microbiota in lambs)

Dear Dr. Shuai Du:

Your manuscript has been accepted, and I am forwarding it to the ASM production staff for publication. Your paper will first be checked to make sure all elements meet the technical requirements. ASM staff will contact you if anything needs to be revised before copyediting and production can begin. Otherwise, you will be notified when your proofs are ready to be viewed.

Sincerely,
Jing Han
Editor
Microbiology Spectrum

Reviewer #2 (Comments for the Author):

The authors has addressed all comments from reviewers.